# Uncertainty-aware Evaluation of Auxiliary Anomalies with the Expected Anomaly Posterior

**Lorenzo Perini**                                       *lorenzo.perini@kuleuven.be*
*Bosch Center for AI, Germany*
*DTAI lab & Leuven.AI, KU Leuven, Belgium*

**Maja Rudolph**                                         *maja.rudolph@us.bosch.com*
*Bosch Center for AI, USA*
*University of Wisconsin-Madison, USA*

**Sabrina Schmedding**                                   *sabrina.schmedding@de.bosch.com*
*Bosch Center for AI, Germany*

**Chen Qiu**                                             *chen.qiu@us.bosch.com*
*Bosch Center for AI, USA*

**Reviewed on OpenReview:** *https://openreview.net/forum?id=Qq4ge9Qe31*

## Abstract

Anomaly detection is the task of identifying examples that do not behave as expected. Because anomalies are rare and unexpected events, collecting real anomalous examples is often challenging in several applications. In addition, learning an anomaly detector with limited (or no) anomalies often yields poor prediction performance. One option is to employ auxiliary synthetic anomalies to improve the model training. However, synthetic anomalies may be of poor quality: anomalies that are unrealistic or indistinguishable from normal samples may deteriorate the detector's performance. Unfortunately, no existing methods quantify the quality of auxiliary anomalies. We fill in this gap and propose the expected anomaly posterior (EAP), an uncertainty-based score function that measures the quality of auxiliary anomalies by quantifying the total uncertainty of an anomaly detector. Experimentally on 40 benchmark datasets of images and tabular data, we show that EAP outperforms 12 adapted data quality estimators in the majority of cases. Code of EAP is available at: `https://github.com/Lorenzo-Perini/ExpectedAnomalyPosterior`.

## 1 Introduction

Anomaly detection aims at identifying the examples that do not conform to the normal behaviour (Chandola et al., 2009). Anomalies are often connected to adverse events, such as defects in production lines (Wang et al., 2024), excess water usage (Perini et al., 2023), failures in petroleum extraction (Martí et al., 2015), or breakdowns in wind turbines (Perini et al., 2022). Detecting anomalies in time can reduce monetary costs and protect resources from harm (Bergmann et al., 2022). For this reason, there has been significant effort to develop data-driven methods for anomaly detection.

Unfortunately, anomalies are inherently rare and sparse, which makes collecting them hard. As a result, the data used to train data-driven methods for anomaly detection only contains a limited number of anomalies. In autonomous driving, for instance, sensor malfunctions or unexpected pedestrian movements are infrequent

but critical to address. As an additional challenge, available anomalies rarely represent all potential cases due to the unpredictable nature of these events. In financial fraud detection, new techniques and schemes are constantly emerging, meaning previously identified anomalies do not cover future fraudulent methods. These factors highlight the difficulty of obtaining comprehensive anomaly datasets, as new and unpredictable anomaly types are inherent to the very nature of these events.

With recent improvements in generative modeling (e.g. (Ho et al., 2020; Dhariwal & Nichol, 2021)) it seems natural to introduce auxiliary anomalies, e.g. for training an anomaly detector (Murase & Fukumizu, 2022), or for model selection (Fung et al., 2023). However, there are several failure cases for generated anomalies that should not be neglected. For one, auxiliary anomalies might be too similar to normal examples. For instance, the defects introduced into images of products might be imperceptible, making the image indistinguishable from a normal counterpart. On the other hand, the quality of a generated anomaly also deteriorates as it becomes too unrealistic, e.g. when the generated defect of a product is too severe. Including poor-quality anomalies for training a model is likely to harm its performance (Hendrycks et al., 2019; Qiu et al., 2022b; Li et al., 2023a;b). Although Chen et al. (2021); Ming et al. (2022) proposed sampling methods for selecting informative anomalies during training, there is no approach for quantifying the quality of auxiliary anomalies.

In this paper, we close this critical gap by introducing the **Expected Anomaly Posterior (EAP)**, the first example-wise score function that measures the quality of auxiliary anomalies. Our approach relies on a fundamental insight: high-quality auxiliary anomalies must fulfill two criteria — they must be (1) distinguishable from normal examples in the training data and (2) realistic, i.e. similar to the training examples (e.g., scratches only affect few pixels, leaving an anomalous image relatively similar to a normal one). Finding a balance between these two characteristics poses an inherent challenge. On one hand, auxiliary anomalies risk deteriorating an anomaly detector's performance if they closely resemble normal examples. On the other hand, they become less useful the more dissimilar from the training data.

Building upon this insight, we adopt a Bayesian framework to model the uncertainty of a detector's prediction. This framework accounts for both an example's dissimilarity from the normal class (via class-conditional probability) and its realism (via example density). More specifically, the class-conditional probability reflects how good a detector is at discriminating the synthetic anomaly from the normal class, while the example density reflects how well the example fits within the overall data distribution. The expectation of the posterior probability that an example is anomalous reflects our concept of the quality of an auxiliary anomaly: the approximation we derive in Section 3 to compute the EAP will give lower scores to indistinguishable and unrealistic anomalies.

In summary, we make three following contributions.

- In Section 3, we compute the expected anomaly posterior (EAP), which measures the quality of an anomaly by accounting both for aleatoric and epistemic uncertainty.

- In Section 4, we provide a theoretical analysis of EAP, including its properties and guarantees.

- In Section 5, we run an extensive experimental analysis and show that EAP enhances a detector's performance when using high-quality anomalies to enrich training or perform model selection.

## 2  Related Work

**Anomaly detection.**  Designing an anomaly detector requires developing a way to assign real-valued anomaly scores to the examples (Chandola et al., 2009; Han et al., 2022b), where the higher the score, the more anomalous the example. Existing approaches often rely on heuristic intuitions about expected anomalous behavior (Pang et al., 2021; Qiu, 2023). Propagation-based detectors, such as those using proximity to training examples, assume similar instances share the same label (e.g., SSDO) (Vercruyssen et al., 2018). Loss-based detectors, on the other hand, learn a decision boundary (e.g., a hypersphere over normals) and assign scores based on the distance to this boundary (Ruff et al., 2020; Zhou et al., 2021; Gao et al., 2021; Qiu et al., 2022a). Self-supervised detectors learn models through solving auxiliary tasks and score anomalies according to model performance on self-supervised tasks (Golan & El-Yaniv, 2018; Bergman & Hoshen, 2020;

Qiu et al., 2021). Recently, foundation models have enabled zero-shot anomaly detection (Jeong et al., 2023), which overcomes the need to collect anomalies for training but still requires their use in model selection (Fung et al., 2023).

**Uncertainty quantification.**   Uncertainty quantification is the task of measuring the reliability of predictions in machine learning models (Hüllermeier & Waegeman, 2021). It typically considers two types of uncertainty: the aleatoric uncertainty, which indicates randomness in the data such as overlapping classes, and the epistemic uncertainty, which reflects the model's lack of knowledge due to limited or biased training data. Bayesian frameworks are commonly used to capture both uncertainties by leveraging their estimated posterior distributions (Bengs et al., 2022). Recent works have introduced advanced methods, such as posterior networks (Charpentier et al., 2020) and second-order scoring rules (Bengs et al., 2023), to refine uncertainty estimates and mitigate limitations in traditional approaches. Such research has shown good results in applied tasks, such as out-of-distribution detection (Ming et al., 2022) and model calibration (Deng et al., 2022), leading to robust tools for tasks that require reliable prediction under uncertainty.

**Data quality.**   Traditional quality score functions evaluate training examples by (1) defining a utility function that takes as input a subset of the training set and measures the performance of the model, and (2) finding a function that assigns a score to an example by quantifying its impact on the model's performance when included/excluded for training (Yoon et al., 2020; Sim et al., 2022; Jiang et al., 2023). Methods like leave-one-out (LOO) iteratively remove one example at a time to observe how test performance varies. Various techniques, such as DATASHAP (Ghorbani & Zou, 2019), BETASHAP (Kwon & Zou, 2021), KNNSHAP (Jia et al., 2019), DATABANZHAF (Wang & Jia, 2023), and AME (Lin et al., 2022), compute the marginal contribution of an example by bootstrapping the training set and assessing its impact on model training. DATAOOB is an out-of-bag (OOB) evaluator that measures out-of-bag accuracy variation. Other methods like LAVA and influence functions (INF) quantify how the utility changes when a specific example is more weighted. The Appendix A.1 provides a detailed overview. Unfortunately, all existing data quality estimators focus on evaluating the impact of training examples and are impractical for evaluating auxiliary anomalies for two reasons. First, retraining a detector multiple times is computationally prohibitive (e.g., LOO). In contrast, EAP trains the detector only once. Second, they require an i.i.d. validation set with the same distribution as the test set to measure the performance variation with and without the example, which is challenging in anomaly detection due to the common distribution gap between validation and test.[1] In contrast, EAP is flexible and can operate effectively even in the absence of a high-quality validation set by directly leveraging the detector's posterior probabilities. This allows it to evaluate auxiliary anomalies without relying on a separate dataset for comparison, which is especially useful in anomaly detection scenarios where a well-matched validation set may not be available. For completeness, we will empirically compare EAP against the main data quality estimators in the experiments (Section 5).

## 3   Methodology

In this Section, we introduce the problem setup and notations (Section 3.1), and describe our proposed approach for quantifying the quality of auxiliary anomalies (Section 3.2).

### 3.1   Problem setup

Let $(\Omega, \mathcal{F}, \mathrm{P})$ be a probability space, and $X\colon \Omega \to \mathbb{R}^d$, $Y\colon \Omega \to \{0, 1\}$ two random variables representing, respectively, feature vectors and class labels (0 for normals, 1 for anomalies).[2] A training dataset is an i.i.d. sample of pairs $D = \{(x_1, y_1), \ldots, (x_n, y_n)\} \sim p(X, Y)$ drawn from the joint distribution. We denote by $\mathrm{P}(Y|X)$ the class conditional probability, and by $p(x)$ the class density. Because of the rarity of anomalies, we assume to have only $m << n$ (labeled) examples from the anomaly class, in addition to $n - m$ (labeled) normal examples.

---

[1]Anomalies may manifest as novelties or out-of-distribution examples, creating distribution shifts on the positive class.
[2]Anomaly detection is often treated as a binary classification task, where anomalies belong to the positive class. See, for example, Chandola et al. (2009) for further information.

Since anomalies provide valuable training signals, but are so rare to acquire, Hendrycks et al. (2019); Murase & Fukumizu (2022) propose using auxiliary anomalies during training. With significant improvements in generative modeling, there are many candidate methods for generating synthetic anomalies to complement the training data. Our goal is to evaluate candidate synthetic anomalies with a quality score function $\phi$ such that higher scores indicate that the synthetic anomaly is useful for training a detector. Before formalizing our research task, we introduce the following definition.

**Definition 3.1** (Categorization of Anomalies). Given the examples $x_R, x_U, x_I \in \mathbb{R}^d$, we define that

- $x_R$ is a *realistic anomaly* if it has high conditional probability and non-zero density

$$P(Y = 1|X = x_R) \in [0.5, 1] \ \text{ and } \ p(x_R) > 0;$$

- $x_U$ is an *unrealistic anomaly* if it has high conditional probability and null density

$$P(Y = 1|X = x_U) \in [0.5, 1] \ \text{ and } \ p(x_U) = 0;$$

- $x_I$ is an *indistinguishable anomaly* if it has null conditional probability and non-zero density

$$P(Y = 1|X = x_I) = 0 \ \text{ and } \ p(x_I) > 0.$$

Strictly speaking, unrealistic anomalies refer to examples falling outside the support of the real data generation model, i.e. with density equal to zero. Note that these are not merely low-density examples, which can be realistic, as anomalies are naturally rare and often occur in low-density regions. On the other hand, indistinguishable anomalies refer to examples that are confidently classified as normals by the anomaly detector, i.e. with class conditional probability of being anomalous equal to zero.

An anomaly detector is a function $f\colon \mathbb{R}^d \to \mathbb{R}$ that assigns a real-valued anomaly score $f(x)$ to any $x \in \mathbb{R}^d$. The detector $f$ is learned using the training set $D$, and can be used for estimating the class conditional probability $P(Y|X)$ by mapping the scores to $[0, 1]$ (Kriegel et al., 2011).

**Given:** $D$ with $m \ll n$ anomalies, a set of $l$ auxiliary anomalies $\{x \in \mathbb{R}^d\}$, and a detector $f$;

**Challenge:** Design a quality score function $\phi\colon \mathbb{R}^d \to \mathbb{R}$ for the auxiliary anomalies, such that for any realistic anomaly $x_R \in l$ and any unrealistic or indistinguishable anomaly $x_U, x_I \in l$, the realistic anomaly has a higher quality score $\phi(x_R) > \phi(x_U), \phi(x_I)$.

With this categorization, estimators for $P(Y|X)$ alone cannot differentiate between $x_R$ and $x_U$, while estimators for $p(X)$ alone cannot differentiate between $x_R$ and $x_I$, thus not qualifying as good quality estimators. Intuitively, a score must quantify the conditional probability to distinguish between $x_R$ and $x_I$ but the estimate needs to account for an example's density. Roughly speaking, the lower an example's density the more uncertain the estimate, because the lack of similar training data prevents a model from learning the correct probability. This introduces an additional level of uncertainty (namely, epistemic), which requires a Bayesian perspective to be properly measured (Hüllermeier & Waegeman, 2021; Bengs et al., 2022).

## 3.2 The Expected Anomaly Posterior

Capturing a detector's uncertainty is challenging because one needs to account for (1) the example's proximity to the normal class (i.e., the aleatoric uncertainty) and (2) the lack of training data in the region where the example falls (i.e., the epistemic uncertainty). This is particularly complicated in anomaly detection because the epistemic uncertainty tends to be high for most anomalies, as they often fall in low-density regions (Bengs et al., 2023).

We propose EAP, a novel approach that estimates the quality of auxiliary anomalies by capturing an anomaly detector's uncertainty. The key idea is to model each example's probability of being an anomaly $\pi_x$. This captures both types of uncertainty: the one on the class prediction $Y \in \{0, 1\}$ (i.e. aleatoric), and the one on the probability $\pi_x$ (i.e. epistemic). The quality score we propose is the expected posterior of this parameter.

**Assumption.** For any $x$ the class conditional distribution $Y|X = x$ is a Bernoulli

$$P(Y|X = x) = \text{BERNOULLI}(\pi_x), \quad \pi_x \sim \text{BETA}(\alpha_0, \beta_0).$$

The parameter $\pi_x$ can be interpreted as the probability of the example $x$ being an anomaly. With the Beta prior we can incorporate prior knowledge such as the expected ratio of anomalies in the data (Perini et al., 2023). Since we have at most one observation of $Y$ for each $x$, we follow Charpentier et al. (2020) and model the posterior over $\pi_x$ by conditioning on pseudo observations. Given $N$ pseudo observations $\bar{y}_1, \ldots, \bar{y}_N$ hypothetically drawn from $P(Y|X = x)$, the posterior,

$$\pi_x|\bar{y}_1, \ldots, \bar{y}_N \sim \text{BETA}(\alpha_0 + \alpha_1, \beta_0 + N - \alpha_1), \tag{1}$$

is conjugate to the Beta prior, where $\alpha_1 = \sum_{i=1}^{N} \mathbf{1}(\bar{y}_i = 1)$ is the number of anomalies in the pseudo observations. That is, if we could sample $N$ labels for the same example $x$, i.e. $(x, y_1), \ldots, (x, y_N)$, we would derive the posterior distribution of $Y|X = x$ by using a simple Bayes update (Eq. 1). However, sampling $N$ labels for the same example $x$ is practically impossible. Thus, we need to parametrize $\alpha_1$. Roughly speaking, if we drew $n$ training examples from $p(X, Y)$ we would expect to draw $N = n \cdot p(x)$ examples with features $x$, among which $P(Y = 1|X = x)$ are anomalies:

$$\alpha_1 \approx n \times \overline{P(Y = 1, X = x)} \approx n \times \underbrace{\overline{P(Y = 1|X = x)}}_{\text{conditional probability}} \times \underbrace{\widehat{p(x)}}_{\text{data density}} \tag{2}$$

where $\hat{\cdot}$ indicates that the quantity is estimated. We describe how we compute both terms below.[3] The expectation of the posterior in Equation (1) reflects the quality of an auxiliary anomaly $x$: if the evidence $N$ is high, we have enough samples to rely on the expected conditional probability for evaluating the auxiliary anomaly (EAP $\approx \overline{P(Y=1|X=x)}$), while if $N$ is low, the quality reflects our prior belief (EAP $\approx \frac{\alpha_0}{\alpha_0 + \beta_0}$). Note that, while $n$ represents the training set size, it can also be treated as a hyperparameter. In this context, it acts like a certainty budget distributed over the two classes.

**Estimating the data density.** Computing $\widehat{p(x)}$ has two main challenges. First, most traditional density estimators suffer the well-known curse of dimensionality (Verleysen & François, 2005; Bengio et al., 2005). Second, deep estimators (e.g., Normalizing Flows (Kobyzev et al., 2020)) are prohibitively time-consuming to be employed for data quality scores. Thus, EAP relies on the rarity score (Han et al., 2022a), which is fast to compute and weakly affected by the curse of dimensionality. The rarity score (1) creates k-NN spheres centered on each training example, and (2) assigns the smallest radius of the sphere that contains the given synthetic example. If the synthetic example falls outside of all spheres, it is considered too uncommon and gets null rarity.

We use the rarity score $r_{\hat{k}}$ with an estimated $\hat{k}$ to estimate the data density.[4] Intuitively, the density behaves as the inverse of the rarity score: highly uncommon examples should have low density. Thus, we take the reciprocal value of the rarity score and normalize it using the training rarity scores:

$$\widehat{p(x)} = \frac{1/r_{\hat{k}}(x)}{1/r_{\hat{k}}(x) + \sum_{i=1}^{n} 1/r_{\hat{k}}(x_i)}. \tag{3}$$

Finally, we want to highlight that the weak estimator in Eq 3 is enough for our task for two reasons. First, assigning null density to synthetic anomalies falling outside of all training spheres allows unrealistic anomalies to be detected. Second, the provided implementation is fast, as one can compute the training spheres only once for the whole set of synthetic examples.

**Estimating the conditional probability.** Computing $\overline{P(Y = 1|X = x)}$ in anomaly detection is a hard task because (1) class probabilities are generally unreliable for imbalanced classification tasks (Wallace &

---

[3] Note that the principle itself is not restricted to this particular choice of estimators. One could apply any other method without altering the validity of the framework.

[4] We explain how we compute $\hat{k}$ in Appendix A.2.

Dahabreh, 2012; Tian et al., 2020), and (2) the available anomalies might be non-representative of the whole anomaly class (i.e., we have access to a biased set). This makes traditional calibration techniques often impractical (Silva Filho et al., 2023; Deng et al., 2022). However, we mainly care about having probabilities that satisfy two properties. First, they must be coherent with the detector's prediction, namely a predicted anomaly (normal) needs a probability greater (lower) than 0.5. Second, we want the proportion of predicted anomalies to match the expected proportion of true anomalies. This guarantees that, if the detector's ranking is accurate, the class predictions are optimally computed.

For this task, we employ a squashing scaler (Vercruyssen et al., 2018) to map the anomaly scores $f(x)$ to $[0, 1]$ probability values

$$\widehat{P(Y = 1|X = x)} = 1 - 2^{-\left(\frac{f(x)}{\lambda}\right)^2} \tag{4}$$

where $\lambda$ is the detector's decision threshold, which we set such that the number of training examples with probability $> 0.5$ (after mapping) is equal to the number of training anomalies $m$ (Perini et al., 2023). Roughly speaking, Eq. 4 uses a monotonic function to map $\lambda$ to 0.5, all scores $> \lambda$ to the interval $(0.5, 1]$, and all the scores $< \lambda$ to $[0, 0.5)$. This guarantees that, when transforming the estimated $\widehat{P(Y = 1|X = x)}$ to predictions by thresholding at 0.5, the predicted classes are the same as when thresholding scores with the decision threshold $\lambda$. This transformation satisfies both properties listed above.

**Computing the quality scores.** Using our point estimates for the data density and the conditional probability, the parametrized posterior distribution $\pi_x|\bar{y}_1, \ldots, \bar{y}_N$ can be computed by substituting to Eq. 1

$$\alpha_1 = n \cdot \widehat{P(Y = 1|X = x)} \cdot \widehat{p(x)} \quad \text{and} \quad N = n \cdot \widehat{p(x)}.$$

Finally, we compute the quality of an auxiliary anomaly $x$ by taking its expectation

$$\phi(x) = \mathbb{E}[\pi_x|\bar{y}_1, \ldots, \bar{y}_N] = \frac{\alpha_0 + n \cdot \widehat{p(x)} \cdot \widehat{P(Y = 1|X = x)}}{\alpha_0 + \beta_0 + n \cdot \widehat{p(x)}}.$$

## 4 Theoretical Analysis of EAP

We theoretically investigate two tasks. First, we illustrate the main properties of EAP, namely how it behaves when subject to (1) large training sets $(n \to +\infty)$, (2) small training sets or zero-density examples, (3) high-class conditional probabilities. Second, we answer the following question: *Given a realistic anomaly $x_R$, an indistinguishable anomaly $x_I$ and an unrealistic anomaly $x_U$ as in Definition 3.1, does EAP rank $\phi(x_R) > \phi(x_I), \phi(x_U)$?*

**i) EAP has three relevant properties:**

*P1) Convergence to class conditional probabilities.* The number of training examples indicates how strong the empirical evidence is. That is, the detector $f$ has enough evidence to estimate properly the class conditional probability. Thus, for high $n$, EAP converges to the class conditional probability

$$\phi(x) \to \widehat{P(Y = 1|X = x)} \qquad \text{for } n \to +\infty;$$

*P2) Convergence to the prior's mean.* No empirical evidence implies that the posterior remains equal to the prior. Thus, EAP assigns the prior's mean for relatively small $n$ or a null-density region,

$$\phi(x) \to \frac{\alpha_0}{\alpha_0 + \beta_0} \qquad \text{for } n \to 0 \ \text{ or } \ \widehat{p(x)} \to 0.$$

*P3) The quality of distinguishable anomalies increases with their density.* Given $\widehat{P(Y = 1|X = x)} \approx 1$ for an example $x$, its quality depends only on its density: the closer/more similar to the training examples, the higher the density, the higher its quality:

$$\phi(x) \approx 1 - \frac{\beta_0}{\alpha_0 + \beta_0 + n \cdot p(x)}.$$

Roughly speaking, assuming that the synthetic anomaly is distinguishable for the anomaly detector from a normal counterpart, the closer it is to the training examples (normal or anomalous) the more likely it will help to improve the decision boundary when used for training.

**ii) EAP' ranking guarantee.**

We show that EAP ranks the anomalies as (1st) realistic, (2nd) unrealistic, and (3rd) indistinguishable.

**Theorem 4.1.** *Let $x_R, x_U, x_I \in \mathbb{R}^d$ be, respectively, a realistic, unrealistic, and indistinguishable anomaly. If the estimators in Eq. 3 and Eq. 4 satisfy the properties of Definition 3.1, then*

$$\frac{\alpha_0}{\alpha_0 + \beta_0} < 0.5 \implies \phi(x_R) > \phi(x_U) > \phi(x_I). \tag{5}$$

*Proof.* Using the definition of indistinguishable and unrealistic anomaly, we immediately conclude

$$\phi(x_U) = \frac{\alpha_0}{\alpha_0 + \beta_0} > \frac{\alpha_0}{\alpha_0 + \beta_0 + n \cdot p(x_I)} = \phi(x_I)$$

because $p(x_I) > 0$. As a second step, we assume that $\frac{\alpha_0}{\alpha_0 + \beta_0} < 0.5$ and show algebraically that

$$\phi(x_R) > \phi(x_U) \iff \phi(x_R) - \phi(x_U) > 0 \iff \frac{\alpha_0 + n \cdot P(Y = 1 | X = x_R) \cdot p(x_R)}{\alpha_0 + \beta_0 + n \cdot p(x_R)} - \frac{\alpha_0}{\alpha_0 + \beta_0} > 0$$

$$\iff n \cdot p(x_R) \left[ (\alpha_0 + \beta_0) \cdot P(Y = 1 | X = x_R) - \alpha_0 \right] > 0 \iff P(Y = 1 | X = x_R) > \frac{\alpha_0}{\alpha_0 + \beta_0},$$

which holds as $P(Y = 1 | X = x_R) > 0.5 > \frac{\alpha_0}{\alpha_0 + \beta_0}$. $\square$

# 5 Experiments

We empirically evaluate two aspects of our method EAP: (a) whether it measures properly the quality of auxiliary anomalies, and (b) its impact on selecting auxiliary anomalies for learning a model or for model selection. For the first aspect, we compare EAP's ability to rank high-quality examples above low-quality ones. For the second aspect, we evaluate the downstream impact of including high-quality samples in training or model selection, measuring how these examples improve the performance of the learned model. We argue that, together, these evaluations illustrate both the standalone utility of EAP as a quality metric and its practical relevance for improving anomaly detection. To this end, we address the following five experimental questions:

Q1. How does EAP compare to existing methods at assigning quality scores?
Q2. How does a model's performance vary when including *high-quality* anomalies for training?
Q3. How does a model's performance vary when including *low-quality* anomalies for training?
Q4. How does the performance of a CLIP-based zero-shot anomaly detection method vary when using the selected auxiliary anomalies for prompt tuning?
Q5. How do EAP' scores vary for different priors?

## 5.1 Experimental Setup

**Baselines.** We compare EAP[5] with 12 adapted baselines: LOO, KNNSHAP (Jia et al., 2019), DATA-BANZHAF (Wang & Jia, 2023), AME (Lin et al., 2022), LAVAEV (Just et al., 2023), INF (Feldman & Zhang, 2020), and DATAOOB (Kwon & Zou, 2023) are existing data quality evaluators that measure the impact of a training example on the model performance. We adapt these methods by including each auxiliary anomaly (individually) in the training set and evaluating its contribution. RANDOMEV assigns uniform random scores to each auxiliary anomaly. RARITY (Han et al., 2022a) computes the rarity score of each auxiliary anomaly. Finally, we include the estimators for the data density $P_x$, the class conditional probability $P_{y|x}$, and a linear combination of them $P_{y|x} + NP_x$.

---

[5]Code is available at: `https://github.com/Lorenzo-Perini/ExpectedAnomalyPosterior`.

**Data.** We carry out our study on 40 datasets, including 15 widely used benchmark image datasets (MvTec) (Bergmann et al., 2019), 3 industrial image datasets for Surface Defect Inspection (SDI) (Wang et al., 2022), and an additional 22 benchmark tabular datasets for anomaly detection with semantically useful anomalies, commonly referenced in the literature (Han et al., 2022b). These datasets vary in size, feature count, and anomaly proportion.

For each dataset, we construct an auxiliary set of $l$ anomalies by combining realistic, indistinguishable, and unrealistic anomalies ($\frac{l}{3}$ each). Realistic anomalies are labeled anomalies provided with the dataset, indistinguishable anomalies are labeled normal examples with swapped labels, and unrealistic anomalies come from other datasets. Specifically, to collect unrealistic anomalies we randomly select 5 datasets out of 40, subsample them to a specific count, and fix their dimensionality to $d$ via random projections (either extending or reducing it). *Pseudo-quality* labels "good" and "poor" are assigned to real anomalies and the other two groups, respectively, reflecting the ground truth where real anomalies should have high-quality scores.

**Setup.** For each dataset, we proceed as follows: (i) We create a balanced test set by adding random normal examples and 50% of available anomalies; (ii) We generate a set of $l$ auxiliary anomalies as described above with $\frac{l}{3} = 40\%$ of available anomalies; (iii) We create a training set by adding 10% of available anomalies and all remaining normal examples to the training set. (iv) We apply all methods to evaluate the external set of anomalies, using the training set for validation when required (as $m \ll n$, we avoid partitioning the training set); To mitigate noise, steps (i)-(iv) are repeated 10 times with different seeds, resulting in a total of 4000 experiments (datasets, methods, seeds). While computing EAP is fast, the baselines have high computational costs because they train a model several times. To run all experiments, we use an internal cluster of six 24- or 32-thread machines (128 GB of memory). The experiments finish in $\sim 72$ hours.

**Models and Hyperparameters.** For all baselines, we use SSDO (Vercruyssen et al., 2018) as the underlying anomaly detector $f$ with $k = 10$ and Isolation Forest (Liu et al., 2008) as prior. This choice is motivated as follows. First, such a combination has been analyzed and used often by researchers (Drogkoula et al., 2023; Stradiotti et al., 2024a; Serban et al., 2024; Pang et al., 2023; Stradiotti et al., 2024b). Second, because some of our datasets include tabular data, we need to employ a fast yet accurate detector for such a data modality. Recent papers such as (Stradiotti et al., 2024b) highlight that SSDO + Isolation Forest is one of the best-performing detectors. When exposed to selected auxiliary anomalies, we employ an SVM with RBF kernel (for images) and a RANDOM FOREST (for tabular data) to make the normal vs. abnormal classification. For images, we use the pre-trained ViT-B-16-SigLIP (Zhai et al., 2023) to extract the features from images and use them as inputs to EAP and all baselines. Our method EAP has one hyperparameter, namely the prior $\alpha_0, \beta_0$, which we set to $\frac{m}{n}$ (the proportion of anomalies in the training set) and $1 - \frac{m}{n}$. Intuitively, this corresponds to the expected proportion of (real) anomalies if an external dataset was sampled from $P(X, Y)$. The baselines[6] have the following hyperparameters: kNNShap and Rarity have $k = 10$, DataBanzhaf, AME, Inf and DataOob use 50 models. All other hyperparameters are set as default (Soenen et al., 2021).

**Evaluation Metrics.** We employ **four** evaluation metrics. First, we use the Area Under the Receiving Operator Curve ($\mathbf{AUC_{qlt}}$) to evaluate the methods' ability to rank good-quality examples higher than poor-quality ones based on quality scores compared to the pseudo-quality labels. Second, we qualitatively analyze the impact of using auxiliary anomalies in training a model, showing the learning curves ($\mathbf{LC_g}$) with the number of added anomalies following the ranking of quality scores on the x-axis and the test accuracy on the y-axis. We compute the area under the learning curve up to $\frac{1}{3}$ of ranked anomalies ($\mathbf{AULC_g}$) and the test accuracy after including top $\frac{1}{3}$ of ranked anomalies ($\mathbf{ACC_g}$). Similarly, we compute the $\mathbf{LC_p}$ following the methods' inverse ordering and measure the $\mathbf{AULC_p}$ of including up to $\frac{2}{3}$ of inversely-ranked anomalies, where lower values are desirable. Finally, separately for each metric, we rank all methods from the best (position 1) to the worst (position 13) in each experiment. By monitoring the rankings, we get insights on "how often" our method outperforms the competitors, as opposed to the traditional "by how much". We denote the ranking-based metrics by adding a $r$ in front in Table 1.

---

[6]Code: `https://github.com/opendataval`

Table 1: Summary of the results obtained by the 13 methods over 18 image (above) and 22 tabular (below) datasets. Columns $6-10$ show the ranking values for the 4 metrics employed (columns $2-5$) and their average. For metrics desiring lower values, we mark with a $\downarrow$. Overall, EAP achieves the best performance and ranking position for most evaluation metrics as well as the best avg. ranking.

**18 Image Datasets**

| Evaluator | $\mathbf{AUC_{qlt}}$ | $\mathbf{AULC_g}$ | $\mathbf{ACC_g}$ | $\mathbf{AULC_p}(\downarrow)$ | $\mathbf{rAUC_{qlt}}$ | $\mathbf{rAULC_g}$ | $\mathbf{rACC_g}$ | $\mathbf{rAULC_p}$ | Avg. Rank |
|---|---|---|---|---|---|---|---|---|---|
| **EAP** | **0.803** | **0.717** | **0.833** | 0.688 | **1.99** | **3.94** | **3.12** | **3.66** | **3.18** |
| Rarity | 0.681 | 0.698 | 0.786 | 0.738 | 4.21 | 5.07 | 5.34 | 7.96 | 5.65 |
| Lava | 0.742 | 0.665 | 0.755 | 0.709 | 2.70 | 8.17 | 7.44 | 4.71 | 5.75 |
| $P_{y|x} + NP_x$ | 0.669 | 0.700 | 0.795 | 0.729 | 4.80 | 5.60 | 5.61 | 7.13 | 5.79 |
| Loo | 0.537 | 0.693 | 0.777 | 0.694 | 7.22 | 5.86 | 6.56 | 6.14 | 6.44 |
| RandomEv | 0.491 | 0.696 | 0.793 | 0.756 | 8.92 | 6.21 | 5.93 | 9.90 | 7.38 |
| DataOob | 0.505 | 0.685 | 0.739 | **0.668** | 8.66 | 7.53 | 9.55 | 4.35 | 7.52 |
| kNNShap | 0.509 | 0.695 | 0.794 | 0.754 | 8.58 | 6.36 | 5.95 | 9.44 | 7.58 |
| AME | 0.512 | 0.693 | 0.793 | 0.756 | 8.56 | 6.42 | 5.85 | 9.94 | 7.69 |
| Inf | 0.488 | 0.681 | 0.778 | 0.744 | 8.87 | 7.01 | 6.73 | 8.55 | 7.79 |
| $P_{y|x}$ | 0.494 | 0.671 | 0.710 | 0.644 | 9.30 | 8.55 | 10.62 | 3.04 | 7.88 |
| DataBanzhaf | 0.500 | 0.668 | 0.775 | 0.748 | 8.71 | 8.01 | 6.68 | 8.54 | 7.98 |
| $P_x$ | 0.502 | 0.535 | 0.600 | 0.733 | 8.59 | 12.37 | 11.72 | 7.72 | 10.10 |

**22 Tabular Datasets**

| Evaluator | $\mathbf{AUC_{qlt}}$ | $\mathbf{AULC_g}$ | $\mathbf{ACC_g}$ | $\mathbf{AULC_p}(\downarrow)$ | $\mathbf{rAUC_{qlt}}$ | $\mathbf{rAULC_g}$ | $\mathbf{rACC_g}$ | $\mathbf{rAULC_p}$ | Avg. Rank |
|---|---|---|---|---|---|---|---|---|---|
| **EAP** | **0.821** | 0.779 | **0.839** | **0.717** | **1.91** | **4.27** | **4.28** | **2.11** | **3.14** |
| Rarity | 0.724 | **0.782** | **0.839** | 0.753 | 3.59 | 4.42 | 4.91 | 4.92 | 4.46 |
| Lava | 0.723 | 0.744 | 0.794 | 0.747 | 3.62 | 7.48 | 7.17 | 4.08 | 5.59 |
| $P_{y|x} + NP_x$ | 0.676 | 0.758 | 0.815 | 0.771 | 4.93 | 6.16 | 6.23 | 6.16 | 5.87 |
| kNNShap | 0.541 | 0.770 | 0.825 | 0.805 | 7.61 | 5.24 | 5.29 | 9.15 | 6.82 |
| RandomEv | 0.502 | 0.773 | 0.827 | 0.809 | 8.25 | 4.92 | 5.33 | 9.72 | 7.06 |
| AME | 0.498 | 0.772 | 0.827 | 0.809 | 8.53 | 5.05 | 5.37 | 9.75 | 7.17 |
| Loo | 0.504 | 0.750 | 0.798 | 0.792 | 7.90 | 6.87 | 7.10 | 7.77 | 7.41 |
| $P_x$ | 0.554 | 0.703 | 0.768 | 0.752 | 6.77 | 10.39 | 9.29 | 4.69 | 7.78 |
| $P_{y|x}$ | 0.533 | 0.712 | 0.748 | 0.753 | 8.20 | 10.56 | 10.66 | 4.30 | 8.43 |
| DataOob | 0.513 | 0.729 | 0.768 | 0.785 | 8.48 | 9.39 | 9.84 | 7.43 | 8.78 |
| Inf | 0.434 | 0.742 | 0.795 | 0.816 | 10.29 | 7.78 | 7.37 | 10.40 | 8.96 |
| DataBanzhaf | 0.415 | 0.736 | 0.789 | 0.817 | 10.92 | 8.47 | 8.16 | 10.53 | 9.52 |

## 5.2 Experimental Results

**Q1. EAP vs baselines at assigning quality scores.** Figure 1 shows the methods' mean $\text{AUC}_{\text{QLT}}$ on both image (left) and tabular data (right). On images, EAP outperforms all baselines on 13 out of 18 datasets, achieving an average $\text{AUC}_{\text{QLT}}$ significantly higher than Lava and Rarity by 6 and 12 percentage points, as shown in Table 1. Also, EAP consistently obtains the lowest average ranking positions (1.99 for $\text{rAUC}_{\text{QLT}}$). On tabular data, EAP obtains an average $\text{AUC}_{\text{QLT}} = 0.821$, which is around 10 percentage points higher than the runner-ups. Also, EAP outperforms Lava and Rarity on 18 and 17 datasets and obtains the best average ranking (1.91).

Interestingly, only EAP, Lava, Rarity, and $P_{y|x} + NP_x$ achieve performance better than random, while other methods get average $\text{AUC}_{\text{QLT}}$ around 0.5, which highlights their inability to distinguish good and poor auxiliary anomalies consistently. As a second remark, EAP performs lower than random for the dataset Tile. This happens because the defective images are extremely different than the normal images, thus resulting in real anomalies falling in zero-density regions, which our method would categorize as unrealistic.

**Q2. Including high-quality anomalies in training.** We measure how the performance of a model varies when introducing the top $\frac{1}{3}$ of auxiliary anomalies into the training following the methods' rankings. Figure 2 (top) shows the learning curves ($\text{LC}_G$) for four representative image datasets. Overall, including high-ranked anomalies first has the claimed impact on the model's performance: the learning curve grows sooner for EAP compared to all baselines. Consequently, EAP obtains an $\text{AULC}_G$ that is, on average, higher than all the baselines by between 2 (vs. Rarity and $P_{y|x} + NP_x$) and 5 (vs. Lava) percentage points. After including one-third of the auxiliary anomalies for training, EAP shows an average improvement on the test performance ($\text{ACC}_G$) of 4 to 10 points over all baselines, standing out as the only method significantly better

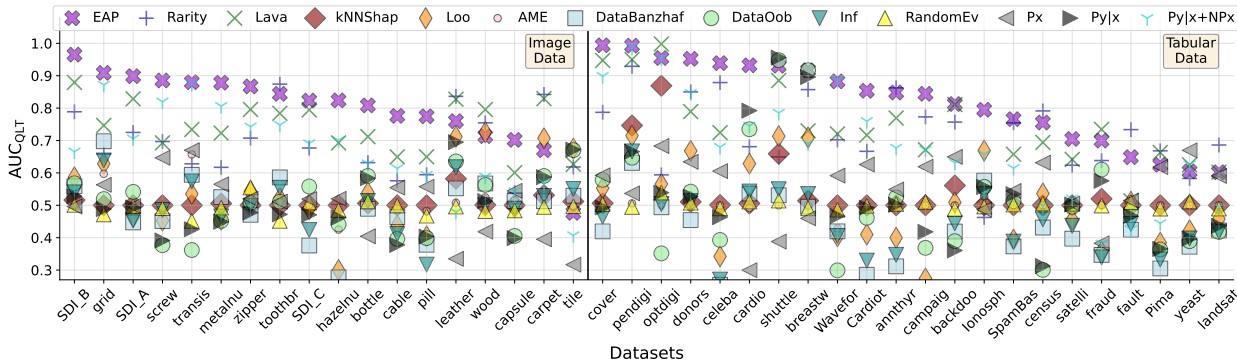

Figure 1: The plot illustrates the average $\text{AUC}_{\text{QLT}}$ obtained by each method on a per-dataset basis (left for image data, right for tabular data). EAP achieves the highest performance for most datasets, beating the runner-ups RARITY and LAVA on, respectively, 30 and 31 datasets out of 40.

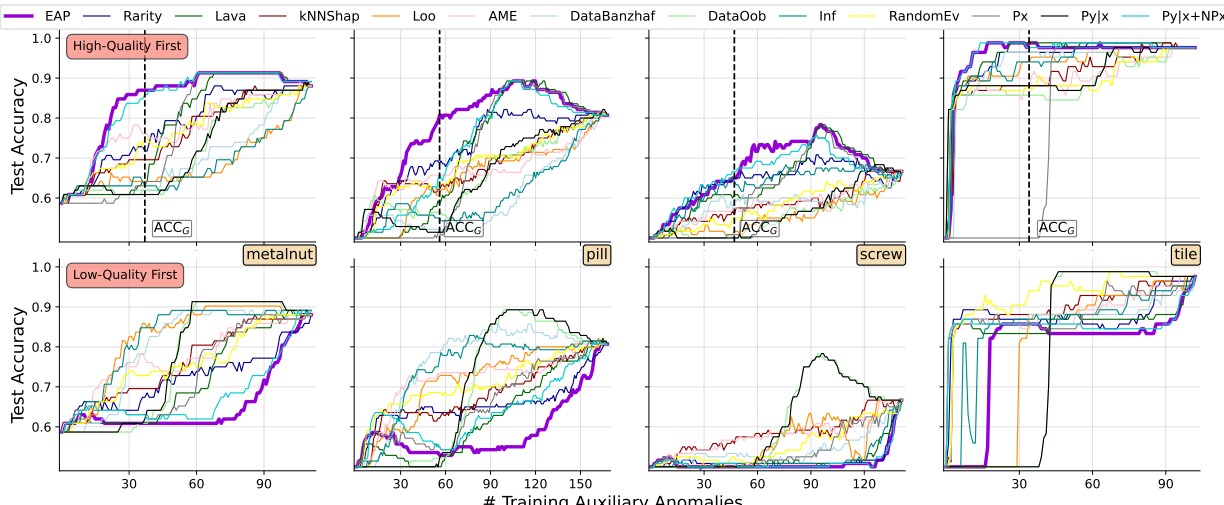

Figure 2: Learning curve (LC) obtained by following the method's ordering (top) and inverse ordering (bottom) for four representative image datasets. Top: EAP' $\text{LC}_{\text{G}}$ grows sooner (i.e., better) than the other methods', which confirms that including high-quality anomalies in the training set has a larger impact on the test performance. Bottom: EAP' $\text{LC}_{\text{P}}$ rises later (i.e., better) than most baselines', showing that low-quality anomalies have a comparatively modest impact on the test performance.

than RANDOMEV. Moreover, Table 1 shows that EAP achieves the best average ranking for both $\text{AULC}_{\text{G}}$ (i.e., 3.94) and $\text{ACC}_{\text{G}}$ (i.e., 3.12).

On tabular data, Table 1 shows that EAP achieves an average $\text{AULC}_{\text{G}}$ slightly lower than RARITY (0.779 vs 0.782) and a similar $\text{ACC}_{\text{G}}$ (both 0.839). This occurs because, for most tabular datasets, RARITY assigns high-quality scores to the unrealistic anomalies which are obviously different than normal data. When including them for training, it yields an improvement of the RANDOM FOREST's test accuracy surprisingly. From a ranking perspective, EAP obtains the best average with $\text{rAULC}_{\text{G}} = 4.27$ (vs. RARITY's 4.42), and $\text{rACC}_{\text{G}} = 4.28$ (vs. RARITY's 4.91).

**Q3. Including low-quality anomalies in training.** Figure 2 (bottom) shows the $\text{LC}_{\text{P}}$ obtained by following an inverse ordering of the methods, i.e., lower ranked anomalies are included first. Using this inverse ordering should result in much slower growth of the $\text{LC}_{\text{P}}$s: in some cases, the test accuracy using EAP remains stable (METALNUT, SCREW), while in others it shows strong fluctuations going up and down quickly (PILL). Interestingly, every baseline's performance goes up for TILE: this is due to their poor ability to assign

Table 2: Test AUCs (%) of prompt tuning with auxiliary anomalies selected by each baseline on MvTec. EAP performs the best on 12 of 15 classes and achieves the highest average AUC.

| Evaluator | bottle | cable | capsule | carpet | grid | hazel | leather | metal | pill | screw | tile | tooth | transis | wood | zipper | avg |
|---|---|---|---|---|---|---|---|---|---|---|---|---|---|---|---|---|
| KNNShap | 82.4 | 56.7 | 51.4 | 82.1 | 58.8 | 72.7 | 92.1 | 51.7 | 49.1 | 60.2 | 69.2 | **77.0** | **77.1** | 72.7 | 44.7 | 66.5 |
| AME | 82.4 | 56.7 | 51.4 | 82.1 | 71.2 | 72.7 | 98.4 | 51.7 | 49.1 | 60.2 | 69.2 | 75.4 | 68.3 | 72.7 | 44.7 | 67.1 |
| LOO | 82.4 | 60.7 | 64.4 | 85.8 | 58.8 | 68.2 | **98.8** | 51.7 | 49.1 | 60.4 | 69.2 | **77.0** | 62.1 | 72.7 | 44.7 | 67.1 |
| DataOob | 82.4 | 60.7 | 63.7 | 85.8 | 71.2 | 72.7 | **98.8** | 51.7 | 49.1 | 60.4 | 69.2 | 75.4 | 62.1 | 72.7 | 44.7 | 68.0 |
| $P_{y|x}$ | 82.4 | 60.6 | 51.2 | 85.8 | 71.2 | 68.2 | 98.4 | 56.2 | 65.6 | 60.4 | 69.2 | 75.4 | 62.1 | 72.7 | 44.7 | 68.3 |
| INF | **95.3** | 56.7 | 45.7 | **96.1** | 83.6 | 68.2 | **98.8** | 51.7 | **65.7** | 60.4 | 69.2 | 75.4 | 62.1 | 93.0 | **75.9** | 73.2 |
| DataBanz | **95.3** | 41.1 | 64.4 | **96.1** | 83.6 | 72.7 | **98.8** | 51.7 | 65.7 | 60.4 | 84.9 | 75.4 | 62.1 | 93.0 | **75.9** | 74.7 |
| RandomEv | **95.3** | 69.2 | 63.7 | **96.1** | 83.1 | 68.2 | **98.8** | 56.2 | **65.7** | 60.4 | 87.8 | 75.4 | 62.1 | **96.4** | **75.9** | 76.9 |
| Rarity | 94.3 | 69.2 | 64.4 | **96.1** | 83.6 | **83.8** | **98.8** | 45.1 | 63.5 | 60.4 | **91.8** | 75.4 | **77.1** | **96.4** | **75.9** | 78.4 |
| $P_x$ | 94.3 | 70.2 | **64.6** | **96.1** | 83.6 | 76.7 | 86.9 | 90.8 | 63.5 | 60.4 | 84.9 | 75.4 | **77.1** | 93.0 | **75.9** | 79.6 |
| Lava | 94.3 | **70.3** | 64.4 | **96.1** | 83.6 | **83.8** | 98.4 | **90.9** | 63.5 | 60.0 | 87.8 | 75.4 | **77.1** | 93.0 | **75.9** | 80.9 |
| $P_{y|x} + NP_x$ | 94.3 | 70.2 | **64.6** | **96.1** | **92.6** | **83.8** | 94.4 | 90.8 | 63.5 | 60.4 | 87.5 | **77.0** | **77.1** | 93.0 | **75.9** | 81.4 |
| **EAP** | **95.3** | **70.3** | 64.4 | **96.1** | **92.6** | **83.8** | **98.8** | **90.9** | 63.5 | **64.8** | **91.8** | **77.0** | **77.1** | 93.0 | **75.9** | **82.4** |

Table 3: Comparison between EAP with default $\alpha_0$ and its six variants $\text{EAP}_w$ that set $\alpha_0 = w, \beta_0 = 1 - w$. Rankings show low sensitivity to such a choice, as long as $\alpha_0 < 0.4$.

| EVALUATOR | $\text{AUC}_{\text{qlt}}$ | $\text{rAUC}_{\text{qlt}}$ | $\text{rACC}_{\text{g}}$ | Avg. Rank |
|---|---|---|---|---|
| EAP | 0.811 | 2.92 | **2.73** | **2.83** |
| $\text{EAP}_{0.2}$ | **0.812** | **2.78** | 3.25 | 3.02 |
| $\text{EAP}_{0.3}$ | 0.810 | 3.39 | 3.40 | 3.40 |
| $\text{EAP}_{0.1}$ | 0.810 | 3.01 | 3.46 | 3.24 |
| $\text{EAP}_{0.05}$ | 0.808 | 3.63 | 3.78 | 3.71 |
| $\text{EAP}_{0.01}$ | 0.805 | 4.10 | 3.54 | 3.82 |
| $\text{EAP}_{0.4}$ | 0.800 | 4.71 | 4.79 | 4.75 |

scores for this dataset, as described in Q1. Surprisingly, DataOob obtains the lowest $\text{AULC}_\text{P}$, while EAP has the second best $\text{AULC}_\text{P}$ with just two percentage points as gap. However, when ranking the experiments, EAP achieves the best average ranking (3.66 of $\text{rAULC}_\text{P}$), thus being the preferred method for most of the experiments.

On tabular data, the results confirm the previous analysis: EAP has the lowest $\text{AULC}_\text{P}$ on 137 experiments out of 220, while Rarity and Lava achieve so only on, respectively, 29 and 31 experiments. This motivates that EAP obtains an average $\text{AULC}_\text{P}$ that is better than all baselines by between 3 and 10 percentage points, as shown in Table 1.

**Q4. Prompt tuning for zero-shot anomaly detection.** CLIP-based anomaly detection methods save the effort of collecting training examples and enable a zero-shot anomaly detection (Jeong et al., 2023). However, their detection performance depends on the choice of prompts, which is usually tuned by using labeled real-world anomalies. We study the impact of selected auxiliary anomalies on prompt tuning for the MvTec datasets. Specifically, we search a prompt for each object class achieving the best performance on the selected auxiliary anomalies from a pool of 27 candidate prompts (see details in Appendix A.3), and apply the best-performing prompt to CLIP at test time.

Table 2 reports the test AUCs of CLIP with the best-performing prompts selected by each data valuation method.[7] We can see that EAP performs the best on 12 of 15 classes and achieves the highest AUC averaged over all classes. Thanks to accounting for both the class conditional probability and the data density, EAP clearly outperforms Rarity and $P_x$, which only consider the data density, $P_{y|x}$, which only considers the class conditional probability, and their naive linear combination $P_{y|x} + NP_x$. The results confirm that EAP selects high-quality auxiliary anomalies for the model selection purpose. We list the prompts selected by EAP in Appendix A.3.

---

[7]Most values are identical because the baselines often select the same prompt from our discrete set of options.

**Q5. EAP' sensitivity to $\frac{\alpha_0}{\alpha_0+\beta_0}$.** EAP requires two hyperparameters: $\alpha_0$, $\beta_0$ of the prior Beta distribution. To remove one degree of freedom, we set $\alpha_0 + \beta_0 = 1$ such that the dataset size $n$ is much stronger ($n$ times) than our initial belief. Then, we investigate how varying the parameter $\alpha_0 \in [0, 0.5)$ impacts EAP's overall performance ($\beta_0 = 1 - \alpha_0$). We compare seven versions of our method by setting $\alpha_0 \in \{0.01, 0.05, 0.1, 0.2, 0.3, 0.4\}$, in addition to the original EAP that leverages the contamination level $\alpha_0 = m/n$. We call $\text{EAP}_w$ the variant that uses $\alpha_0 = w$. Table 3 shows the rankings of these 7 variants for $\text{AUC}_{\text{QLT}}$ and $\text{ACC}_{\text{G}}$ and their average.[8] Overall, both parameters have a low impact on our method: while a higher value for $\alpha_0$ improves the $\text{AUC}_{\text{QLT}}$, in some experiments this improvement does not yield better performance at test time in terms of $\text{ACC}_{\text{G}}$. Moreover, setting $\alpha_0$ too high or too low has inherent risks: $\text{EAP}_{0.4}$ and $\text{EAP}_{0.01}$ are the worst variants by far, with significant drops in performance compared to the other variants.

## 6 Conclusion

This paper addressed the problem of evaluating the quality of an auxiliary set of synthetic anomalies. With this quality score, one can enrich an anomaly detection dataset to learn a more accurate anomaly detector. We proposed the expected anomaly posterior (EAP), the first quality score function for auxiliary anomalies derived from an approximation for the posterior over the probability that a given input is an anomaly. We showed that our approach theoretically assigns higher scores to the realistic anomalies, compared to unrealistic and indistinguishable anomalies. Empirically, we investigated how EAP compares to adapted data quality estimators at (1) assigning quality scores, (2) using such scores to enrich the data for training, and (3) model selection. On 40 datasets, we show that EAP outperforms all 12 baselines in the majority of the cases.

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

# A   Appendix

## A.1   Data quality estimators.

Any data quality estimator can be seen as a mapping that assigns a scalar score to any example $(x, y)$. Such a score quantifies the impact of $(x, y)$ on the model's performance when trained including the example in the training set. For this task, they introduce a utility function $U(\bar{D}) \coloneqq \mathrm{PERF}(f, \bar{D})$ that takes as input a subset $\bar{D}$ of $D$ and measures the performance of $f$ when trained on it. Next, we briefly describe the existing data quality estimators employed in the experiments and refer to (Jiang et al., 2023) for additional details.

- LEAVE ONE OUT (LOO) is defined as $\phi_{\mathrm{LOO}}(x, y) = U(D) - U(D \backslash \{(x, y)\})$, where $U$ is commonly chosen as the accuracy;
- DATASHAP generalizes LOO's approach to the concept of marginal contributions, which measures the average change in utility when $(x, y)$ is removed from any training set. Given a training set cardinality $j \leq N$, the marginal contribution is defined as

$$\mathcal{M}_j(x, y) \coloneqq \binom{N-1}{j-1}^{-1} \sum_{\bar{D}_j \subseteq D, |\bar{D}_j| = j-1} U(\bar{D}_j \cup \{(x, y)\}) - U(\bar{D}_j)$$

where $\bar{D}_j$ is a random subset of $D$ of cardinality $j-1$ that does not contain $(x, y)$. Then, DATASHAP (Ghorbani & Zou, 2019) computes the score as $\phi_{\mathrm{DATASHAP}}(x, y) = \frac{1}{N} \sum_{j=1}^{N} \mathcal{M}_j(x, y)$;
- BETASHAP (Kwon & Zou, 2021) generalizes DATASHAP by considering a weighted average of marginal contributions $\phi_{\mathrm{BETASHAP}}(x, y) = \frac{1}{N} \sum_{j=1}^{N} \omega_j \mathcal{M}_j(x, y)$, for some weights $\omega_1, \ldots, \omega_N$.
- DATABANZHAF (Wang & Jia, 2023) exploits the same formulation as BETASHAP but sets the weights to $\omega_j = 2^{-N} \binom{N-1}{j-1}$.
- AME (Lin et al., 2022) shows that the average marginal contribution taken over random subsets of $D$ can be efficiently estimated by predicting the model's prediction. They employ a LASSO regression model that minimizes

$$\underset{\gamma \in \mathbb{R}^N}{\arg\min} \, \mathbb{E} \left[ U(\bar{D}) - g(\mathbb{1}(\bar{D}))^T \gamma \right]^2 + \lambda \sum_{i=1}^{N} |\gamma_i|,$$

where $\mathbb{1}(\bar{D})$ is the multi-dimensional characteristic function, $\bar{D}$ is a random subset draw the data distribution, $\lambda$ is the regularization parameter, and $g \colon \{0, 1\} \to \mathbb{R}^N$ is a predefined transformation. The values $\gamma_i$ represent the quality of $(x_i, y_i)$.
- KNNSHAP (Jia et al., 2019) differs from DATASHAP on the choice of the utility function:

$$U(\bar{D}) = \frac{1}{N_{val} k} \sum_{i=1}^{N_{val}} \sum_{(x_j, y_j) \in \mathcal{N}(x_i, \bar{D})} \mathbb{1}(y_i = y_j),$$

where $k$ is the number of neighbors, $N_{val}$ is the size of the validation set, and $\mathcal{N}(x_i, \bar{D})$ indicates the set of nearest neighbors for the validation example $x_i$ over the subset $\bar{D}$. Roughly speaking, it measures the proportion of examples in $\bar{D}$ that are neighbors of $x_i$ and share the same label $y_i$.
- INFLUENCE FUNCTIONS (INF) (Feldman & Zhang, 2020) approximate the difference of utility functions in LOO by splitting $D$ into two subsets of equal cardinalities and randomly drawing subsets from each of them:

$$\phi_{\mathrm{INF}}(x, y) = \mathbb{E}_{\bar{D}_x}[U(\bar{D}_x)] - \mathbb{E}_{\bar{D}_{\not{x}}}[\bar{D}_{\not{x}}],$$

where all the subsets from $\bar{D}_x$ contain $(x, y)$, while none of the subsets from $\bar{D}_{\not{x}}$ contain $(x, y)$.
- LAVA (Just et al., 2023) measures the quality of $(x, y)$ by quantifying how fast the optimal transport cost between the training and validation sets changes when increasing more weight to $(x, y)$. That is,

$$\phi_{\mathrm{LAVA}}(x, y) = h^* - \frac{1}{N-1} \sum_j h_j^*,$$

where $h_i^*$ is part of the optimal solution of the transport problem.

- DataOob (Kwon & Zou, 2023) relies on the concept of out-of-bag estimate to capture the data quality. Given $B$ weak learners $f_b$, each trained on a bootstrap sample of $D$, the quality score is

$$\phi_{\text{DataOob}}(x, y) = \frac{\sum_{b=1}^{B} \mathbb{1}(w_b = 0) T(y, f_b(x))}{\sum_{b=1}^{B} \mathbb{1}(w_b = 0)},$$

where $w_b$ is the number of times $(x, y)$ is selected in the $b$−th bootstrap, and $T$ is an evaluation metric (e.g., correctness $\mathbb{1}(y = f_b(x))$).

## A.2 Rarity score

Formally, given a synthetic image with extracted feature $x$, the rarity score is a function $r_k \colon \mathbb{R}^d \to \mathbb{R}$ such that

$$r_k(x) = \begin{cases} 0 & \text{if } x \notin \bigcup_{x_i \in D} B_k(x_i) \\ \min_{x_i \in D \colon x \in B_k(x_i)} NN_k(x_i) & \text{otherwise} \end{cases} \tag{6}$$

where $NN_k(x_i)$ is the distance between $x_i$ and its k-th nearest neighbor in $D$, and $B_k(x_i) = \{x | d(x_i, x) \leq NN_k(x_i)\}$ is the k-NN sphere with $x_i$ as center and $NN_k(x_i)$ as radius. The rarity score strongly depends on the choice of the hyperparameter $k$: high values of $k$ could map far unrealistic examples to a positive high score, namely they would be considered authentic, while low values of $k$ could map real examples slightly different than the training data to a null score, namely they would be considered artifacts.

Because the rarity score strictly depends on the hyperparameter $k \in \{1, \ldots, n-1\}$, we need to estimate a proper value $\hat{k}$. Let's assume the existence of an optimal $k$, and use the small set of $m$ anomalies to estimate it. Ideally, $k$ should be: (1) as low as possible to assign null scores to unrealistic anomalies, and (2) high enough to assign positive scores to the real training anomalies.

Following this insight, we assume a Bayesian perspective and set a normalized variable's $K$ prior to be uniform

$$K := \frac{k-1}{n-1} \sim \text{Beta}(1, 1) = \text{Unif}(0, 1).$$

Roughly speaking, we min-max normalize $K$ to $[0, 1]$ to exploit that a Beta prior with a Bernoulli likelihood results in a Beta posterior distribution. Because we want the minimum $k$ that assigns positive scores to the training anomalies $\{x_{\bar{m}}\}_{\bar{m} \leq m}$, we compute for each $x_{\bar{m}}$ the minimum $k_{\bar{m}}$ such that $r_{k_{\bar{m}}}(x_{\bar{m}}) > 0$. The set of normalized $\{\frac{k_{\bar{m}} - 1}{n-1}\}_{\bar{m} \leq m}$ is the empirical evidence for the Bayesian update, which is

$$K \Big| \Big\{ \frac{k_{\bar{m}} - 1}{n-1} \Big\} \sim \text{Beta}\left( 1 + \sum_{\bar{m} \leq m} \frac{k_{\bar{m}} - 1}{n-1}, 1 + m - \sum_{\bar{m} \leq m} \frac{k_{\bar{m}} - 1}{n-1} \right).$$

Finally, we estimate $\hat{k}$ as the 95th percentile of the posterior distribution of $K$

$$\hat{k} = \arg\min_{t \in [0,1]} \mathbb{P}\left( K \Big| \Big\{ \frac{k_{\bar{m}} - 1}{n-1} \Big\} \leq t \right) \geq 0.95 \tag{7}$$

which guarantees that at least 95% of real anomalies get a positive rarity score.

## A.3 Prompt tuning for CLIP

```
%Candidate prompt templates for MvTec:
    ['{}','damaged {}'],
    ['flawless {}','{} with flaw'],
    ['perfect {}','{} with defect'],
    ['unblemished {}','{} with damage'],
    ['{} without flaw','{} with flaw'],
    ['{} without defect','{} with defect'],
    ['a photo of a normal {}','a photo of an anomalous {}'],
    ['a cropped photo of a normal {}', 'a cropped photo of an anomalous {}'],
    ['a dark photo of a normal {}', 'a dark photo of an anomalous {}'],
    ['a photo of a normal {} for inspection', 'a photo of an anomalous {} for inspection'],
    ['a photo of a normal {} for viewing', 'a photo of an anomalous {} for viewing'],
    ['a bright photo of a normal {}', 'a bright photo of an anomalous {}'],
    ['a close-up photo of a normal {}', 'a close-up photo of an anomalous {}'],
    ['a blurry photo of a normal {}', 'a blurry photo of an anomalous {}'],
    ['a photo of a small normal {}', 'a photo of a small anomalous {}'],
    ['a photo of a large normal {}', 'a photo of a large anomalous {}'],
    ['a photo of a normal {} for visual inspection', 'a photo of an anomalous {} for visual inspection'],
    ['a photo of a normal {} for anomaly detection','a photo of an anomalous {} for anomaly detection'],
    ['a photo of a {}','a photo of something'],
    ['a cropped photo of a {}', 'a cropped photo of something'],
    ['a dark photo of a {}', 'a dark photo of something'],
    ['a photo of a {} for inspection', 'a photo of something for inspection'],
    ['a bright photo of a {}', 'a bright photo of something'],
    ['a close-up photo of a {}', 'a close-up photo of something'],
    ['a blurry photo of a {}', 'a blurry photo of something'],
    ['a photo of a {} for visual inspection', 'a photo of something for visual inspection'],
    ['a photo of a {} for anomaly detection','a photo of something for anomaly detection']
```

```
%EAP selected prompts for MvTec:
    ['bottle', 'damaged bottle'],
    ['cable without defect', 'cable with defect'],
    ['unblemished capsule', 'capsule with damage'],
    ['carpet', 'damaged carpet'],
    ['a bright photo of a normal grid', 'a bright photo of an anomalous grid'],
    ['hazelnut without defect', 'hazelnut with defect'],
    ['a photo of a normal leather for inspection', 'a photo of an anomalous leather for inspection'],
    ['metalnut', 'damaged metalnut'],
    ['pill', 'damaged pill'],
    ['a close-up photo of a screw', 'a close-up photo of something'],
    ['tile', 'damaged tile'],
    ['toothbrush without flaw', 'toothbrush with flaw'],
    ['a blurry photo of a normal transistor', 'a blurry photo of an anomalous transistor'],
    ['wood', 'damaged wood'],
    ['zipper without defect', 'zipper with defect']
```

Table 4: Summary of the results obtained by the 13 methods over all 40 datasets. We report the mean $\pm$ std, computed over all experiments. Overall, EAP achieves the best performance and ranking position for all evaluation metrics as well as the best average ranking (last column).

| EVALUATOR | $\mathbf{AUC_{qlt}}$ | $\mathbf{AULC_g}$ | $\mathbf{ACC_g}$ | $\mathbf{AULC_p(\downarrow)}$ | $\mathbf{rAUC_{qlt}}$ | $\mathbf{rAULC_g}$ | $\mathbf{rACC_g}$ | $\mathbf{rAULC_p}$ | Avg. Rank |
|---|---|---|---|---|---|---|---|---|---|
| **EAP** | $\mathbf{0.81 \pm 0.13}$ | $\mathbf{0.76 \pm 0.15}$ | $\mathbf{0.84 \pm 0.14}$ | $\mathbf{0.70 \pm 0.14}$ | $\mathbf{1.95 \pm 1.65}$ | $\mathbf{4.12 \pm 2.88}$ | $\mathbf{3.76 \pm 2.75}$ | $\mathbf{2.82 \pm 2.17}$ | $\mathbf{3.16 \pm 1.67}$ |
| RARITY | $0.70 \pm 0.14$ | $0.74 \pm 0.16$ | $0.82 \pm 0.16$ | $0.75 \pm 0.15$ | $3.87 \pm 2.73$ | $4.71 \pm 3.36$ | $5.11 \pm 3.51$ | $6.29 \pm 3.49$ | $4.99 \pm 2.68$ |
| LAVA | $0.73 \pm 0.13$ | $0.71 \pm 0.16$ | $0.78 \pm 0.17$ | $0.73 \pm 0.14$ | $3.21 \pm 2.03$ | $7.79 \pm 3.63$ | $7.29 \pm 3.62$ | $4.36 \pm 2.54$ | $5.66 \pm 2.19$ |
| $P_{y|x} + NP_x$ | $0.67 \pm 0.18$ | $0.73 \pm 0.16$ | $0.81 \pm 0.16$ | $0.75 \pm 0.14$ | $4.82 \pm 3.22$ | $5.87 \pm 3.40$ | $5.90 \pm 3.37$ | $6.55 \pm 3.22$ | $5.79 \pm 2.53$ |
| LOO | $0.52 \pm 0.17$ | $0.72 \pm 0.16$ | $0.79 \pm 0.16$ | $0.75 \pm 0.14$ | $7.59 \pm 3.48$ | $6.42 \pm 3.57$ | $6.86 \pm 3.31$ | $7.04 \pm 3.42$ | $6.98 \pm 2.90$ |
| KNNSHAP | $0.53 \pm 0.08$ | $0.74 \pm 0.16$ | $0.81 \pm 0.15$ | $0.78 \pm 0.15$ | $8.05 \pm 2.02$ | $5.74 \pm 2.84$ | $5.59 \pm 2.60$ | $9.28 \pm 2.36$ | $7.16 \pm 1.40$ |
| RANDOMEV | $0.50 \pm 0.06$ | $0.74 \pm 0.16$ | $0.81 \pm 0.15$ | $0.79 \pm 0.15$ | $8.55 \pm 2.43$ | $5.50 \pm 2.63$ | $5.60 \pm 2.45$ | $9.80 \pm 2.21$ | $7.36 \pm 1.66$ |
| AME | $0.50 \pm 0.05$ | $0.74 \pm 0.16$ | $0.81 \pm 0.15$ | $0.79 \pm 0.15$ | $8.54 \pm 2.06$ | $5.66 \pm 2.59$ | $5.58 \pm 2.50$ | $9.84 \pm 2.17$ | $7.41 \pm 1.47$ |
| $P_{y|x}$ | $0.52 \pm 0.14$ | $0.69 \pm 0.16$ | $0.73 \pm 0.16$ | $\mathbf{0.70 \pm 0.13}$ | $8.69 \pm 3.10$ | $9.66 \pm 3.07$ | $10.64 \pm 2.46$ | $3.73 \pm 2.67$ | $8.18 \pm 2.13$ |
| DATAOOB | $0.51 \pm 0.15$ | $0.71 \pm 0.15$ | $0.75 \pm 0.16$ | $0.73 \pm 0.14$ | $8.56 \pm 3.38$ | $8.55 \pm 3.15$ | $9.71 \pm 2.56$ | $6.04 \pm 3.71$ | $8.22 \pm 2.44$ |
| INF | $0.46 \pm 0.14$ | $0.71 \pm 0.16$ | $0.79 \pm 0.17$ | $0.78 \pm 0.14$ | $9.65 \pm 2.61$ | $7.43 \pm 3.17$ | $7.08 \pm 2.95$ | $9.56 \pm 2.79$ | $8.43 \pm 2.26$ |
| $P_x$ | $0.53 \pm 0.11$ | $0.63 \pm 0.15$ | $0.69 \pm 0.16$ | $0.74 \pm 0.14$ | $7.59 \pm 3.59$ | $11.28 \pm 2.68$ | $10.38 \pm 3.32$ | $6.06 \pm 3.27$ | $8.83 \pm 2.51$ |
| DATABANZHAF | $0.45 \pm 0.14$ | $0.71 \pm 0.16$ | $0.78 \pm 0.17$ | $0.79 \pm 0.14$ | $9.92 \pm 2.89$ | $8.26 \pm 3.24$ | $7.49 \pm 3.24$ | $9.63 \pm 2.72$ | $8.83 \pm 2.39$ |

