# OpenReview forum: "Uncertainty-aware Evaluation of Auxiliary Anomalies with the Expected Anomaly Posterior"
_TMLR — Accepted by TMLR_

### Review · Reviewer_MpzE · 2024-11-20

**Summary Of Contributions:**

This paper introduces an uncertainty-aware scoring function for evaluating the quality of auxiliary anomalies. By integrating class-conditional probability and data density, the proposed method addresses the challenge of assessing synthetic anomaly quality while enhancing anomaly detector performance. Both theoretical analysis and extensive experiments are provided.

**Audience:**

Yes

**Claims And Evidence:**

Yes

**Requested Changes:**

see above

**Strengths And Weaknesses:**

Pros:
1. Anomaly quality evaluation is a previously unexplored problem.
2. Extensive evaluation validates the robustness and versatility of the proposed method.
3. EAP improves both training and model selection.

Cons:
1. The problem of anomaly quality evaluation is intriguing and novel, as stated in the paper. However, after analyzing the method and experiments, it seems that the core task is still anomaly detection rather than evaluating anomaly quality. The design and experiments essentially focus on improving the performance of an anomaly detector $f$, which makes the contribution feel somewhat overstated compared to the high-level goal.
2. In the contribution section, the paper first introduces aleatoric and epistemic uncertainty, which are central to the proposed approach. However, these concepts are not adequately explained, making it hard for readers unfamiliar with the background to follow. Additionally, while the introduction presents the high-level challenges (balancing dissimilarity and realism), the paper lacks sufficient technical details on how these challenges are addressed, making me confuse with the underlying mechanisms.
3. The algorithm is built on two key aspects: dissimilarity and realism. However, the paper does not clearly explain why conditional probability is an appropriate metric for dissimilarity or why data density represents realism (as suggested in Eqs 2 and 3). Without a rigorous justification, these choices seem arbitrary, reducing the clarity and confidence in the proposed framework.
4. The estimation of rarity $r_k$ and the selection of the anomaly detector $f$ are not well justified. For instance, the paper uses SSDO and Isolation Forest in experiments but does not explain the rationale behind these choices. If the method is truly generalizable, it should also be tested with more advanced anomaly detection methods. The lack of exploration into whether the proposed method is detector-agnostic leaves room for skepticism about its universality.
5. Experiments: (1) In Table 2, certain results are identical across methods for some datasets, why? (2)  Including visualizations of normal and synthetic images, particularly for the three types of anomalies (realistic, indistinguishable, and unrealistic), would greatly aid understanding. (3) The spacing between figures in the experimental section is too small, making the paper difficult to read and visually unappealing.

---

> ### Author Response · Authors · 2024-12-12
> **Response to reviewer MpzE - Part 1**
>
> Dear reviewer MpzE,
>
> We appreciate your thoughtful review and constructive feedback. We have fixed the text and marked all changes in orange.
>
> Below, we address the points raised in detail:
>
> - [**Core Task: Anomaly Detection vs. Quality Evaluation**] While improving anomaly detection performance is an essential goal, our primary focus is on evaluating synthetic anomaly quality. We argue that these are closely intertwined: high-quality auxiliary anomalies significantly improve the performance of anomaly detectors when used for model training. Our proposed metric, EAP, is empirically evaluated from two complementary perspectives. First, as shown in Figure 1 and Table 1, we compare its ability to rank high-quality examples above low-quality ones. This assessment directly demonstrates the metric's capability to distinguish good-quality from poor-quality anomalies. Second, in Figure 2 and Table 2, we evaluate the downstream impact of including high-quality samples in training, measuring how these samples improve the performance of the trained model. We argue that, together, these evaluations illustrate both the standalone utility of EAP as a quality metric and its practical relevance for improving anomaly detection. We include this discussion at the beginning of Sec 5 of the paper.
>
> - [**Explanation of Uncertainty Concepts**] Aleatoric and epistemic uncertainties are central to understanding our approach, and we will expand their descriptions with accessible explanations and references. Specifically, the aleatoric uncertainty reflects class overlap and is captured by $P(Y|X)$, while the epistemic uncertainty reflects the model’s confidence, which depends on how similar a sample is to the training instances and is captured by $p(X)$. We include a brief description in the related work.
>
> - [**Justification of Conditional Probability and Density**] Motivated by the principles of dissimilarity and realism, we leverage class conditional probability and data density to quantify the quality of anomalies. The class conditional probability $P(Y=1|X)$ reflects how good a detector is at discriminating the synthetic anomaly from the normal class, capturing the dissimilarity aspect of our framework. Meanwhile, the example density $p(X)$ reflects how well the synthetic anomaly fits within the overall data distribution (i.e. if it falls in low-density regions or null-density regions). Anomalies closer to high-density regions are more plausible and thus realistic. However, such anomalies may not always be easily distinguishable. Our framework leverages both metrics to balance these competing aspects. We will include this discussion and additional clarification in the Introduction of the paper.
>
> - [**Estimation of Rarity Score**] The rarity score $r_k$ serves as a simple yet effective proxy for density estimation and is aligned with our goal of general applicability across data modalities (tabular, images) and domains. Traditional density estimators like KDE are unsuitable for high-dimensional data (e.g., images) while network-based estimators are computationally expensive and infeasible to use within our large experimental comparison, as discussed in the paper (Sec 3.2 - data density).
>
> - [**Choice of SSDO and IForest**] Our choice of SSDO and Isolation forest is motivated by two aspects. First, such a combination has been analyzed and used often by researchers [2,3,4,5,6]. Second, because some of our datasets include tabular data, we need to employ a fast yet accurate detector for such a data modality. Recent papers such as [6] highlight that SSDO+IsolationForest is one of the best-performing detectors that are not network-based (i.e., does not require long training time). We did not directly include the approach in [6] as its computational cost is close to a network-based model. We acknowledge that testing the proposed metric with different anomaly detectors could provide additional insights. However, we believe this is not essential to demonstrate the effectiveness of our metric EAP. Our primary goal is to evaluate the quality of synthetic anomalies, and we argue that the design choices for the density estimators and anomaly detectors used in our experiments are both simple and standard. This ensures that the results are generalizable and not overly reliant on specific detector architectures. Adding experiments with varying detectors would introduce additional variability and noise, possibly hiding key insights. We hope the reviewer will understand and agree that our current experimental setup sufficiently supports the evaluation of our proposed metric.

---

> > ### Author Response · Authors · 2024-12-12
> > **Response to reviewer MpzE - Part 2**
> >
> > - [**Experiments**] First, most of the results in Table 2 are identical because some of the baselines pick the same prompt when doing prompt tuning with a finite (discrete) list of candidate prompts. Second, as described in Section 5.1 (Data), the synthetic dataset for evaluating our metric is constructed by using real anomalies as realistic anomalies, real normal samples as indistinguishable anomalies, and samples from other irrelevant classes as unrealistic anomalies. While visualizations can be valuable, we believe the text sufficiently explains the dataset creation process, particularly given the space constraints in the main text. Instead, we will extend the text to emphasize these distinctions more explicitly to enhance clarity. Third, we remove one of the plots from Fig.2 to leave more space, as suggested.
> >
> > [2] Drogkoula M, Kokkinos K, Samaras N. A comprehensive survey of machine learning methodologies with emphasis in water resources management. Applied Sciences, 2023.
> >
> > [3] Stradiotti L, Perini L, Davis J. Combining Active Learning and Learning to Reject for Anomaly Detection. ECAI 2024.
> >
> > [4] Serban CM, Sebestyen G, Hangan A. Anomaly Detection in Water Consumption Patterns Using Prediction and Clustering Approaches. IEEE, 2024.
> >
> > [5] Pang G, Shen C, Jin H, van den Hengel A. Deep weakly-supervised anomaly detection. SIGKDD, 2023.
> >
> > [6] Stradiotti L, Perini L, Davis J. Semi-Supervised Isolation Forest for Anomaly Detection. In Proceedings of the 2024 SIAM International Conference on Data Mining (SDM) 2024.

---

### Review · Reviewer_FR6C · 2024-11-26

**Summary Of Contributions:**

The paper proposes a method to rank auxiliary anomalies, namely artificially generated anomalies that can be used to train anomaly detectors. An anomaly is defined as a dataset sample that does not conform to the normal behaviour, e.g. defects in production lines. Detecting such anomalies can be essential in numerous industrial applications, and adding auxiliary anomalies to the training set of anomaly detectors could make a difference in improving their representation since they are - by definition - low density samples.

The key innovation is the use of the expected anomaly posterior as the quality score of auxiliary anomalies.
The authors assume the probability of each sample being an anomaly to be modelled as a a Bernoulli distribution over the examples. One sample is either an anomaly or it is not. The authors then model the probability of drawing an anomaly out of $N$ draws $(N = n \cdot p(x))$ of the same features $x$ as the conditional probability $P(Y=1|X=x)$, for which they can compute the expectation when $N$ is high. If $N$ is low, the value can be then formulated in relationship to a prior representing the expected ratio of anomalies in the data.
The expectation of this conditional probability is used as the quality score.

Further contributions are the estimation of the data density and conditional probability, which are necessary to get to the quality score. The data density is estimated based on the inverse of the rarity score. The conditional probability is then computed by a squashing scaler over the anomaly scores.

**Audience:**

Yes

**Broader Impact Concerns:**

No ethical statement was required for this work.

**Claims And Evidence:**

Yes

**Requested Changes:**

I would like to suggest to the authors some improvements:
- The paper is at times written unclearly or hard to follow. I detail these parts below, where the authors could improve the clarity by rephrasing some sentences or giving explicit examples.
- In section 3.2, the concept of sampling $N$ labels for the same example $x$ (virtually by assigning a probability distribution to it) could be illustrated with a figure, as this would make the concept clearer for the reader
- Eq. 3 is based on a strong assumption that the rarity score can be used to estimated data density. I think the authors should add some basic experiments to strengthen this claim where they compare traditional density estimation to this method and assess formally the error. It would make the contribution stronger.
- In Eq. 4 there is a $f(x)$ that had not been defined elsewhere. For me it was a bit confusing, because I do not think that there should be $f(x)$ there. Maybe the authors could clarify or correct the equation, if needed.
- The authors should comment on P3 in page 6. Essentially, their method strongly depends on the density of observed anomalies. However, they justified the method by stating that anomalies are -by definition- sparsely distributed and that this is why auxiliary anomalies are needed as a form of data augmentation. I see a trade-off here that could have been discussed further, because this could be a main limitation of the method. It would be nice to see how by varying the density of anomalies the results remain stable.
- This point links to the one above. The authors assigned $l/3$ for each type of anomalies when building the dataset. This fraction shows only one side of the medal. How would the method perform when, for instance, no realistic anomalies are available (or only an infinitesimal fraction is, e.g. $l/1000$)
- In Q5 on page 10, the authors evaluate the sensitivity to the hyper parameters. I was disappointed to see the result table in the appendix instead of the main paper, so I would move it, if possible. Moreover, it is not clear to me why only the rAUC was used in the evaluation. Would it be possible to add the AUC values as well?


Sections where the writing was not clear:
- On page 7 the authors use a lot of terms in parentheses. I would avoid because it becomes very confusing to follow. For example, when they state "EAP consistently obtains the lowest (best) average .. " I would choose only one term between the two. Similarly on the caption of Table 1 (page 8) and on page 9.
- In "evaluation metrics" I really struggled to understand what the rankings score rAUC meant. Since the authors use this metric in several places to describe the improvement of their method over the baselines, the authors should clarify what this means by giving an example or making the description less cryptic.

**Strengths And Weaknesses:**

Overall, I am positive for this paper being accepted in TMLR (but some changes should be made). The authors did a great job in formulating the method and the experiments are extensive and well supported. Their contribution in the formulation of a quality score is relevant and their use of the expected anomaly posterior is novel and interesting. The results compare the method to a variety of baselines and show a consistent improvement.

Below a list of strengths:
- the results show evidence of the method's improvement
- the quality scores are well formulated and cheap to compute
-  the impact of adding the auxiliary samples during training has been investigated and documented
- sensitivity to hyperparameter tuning has been discussed

My main concern while reading this paper has been the motivation. I recognised that this may span from a lack of expertise in the matter from my side, although I have been working on uncertainty estimation for a while. Is it really needed to model anomaly detection as a binary classification task? I see anomalies as marked outliers, something that can be identified as shifting from the original data distribution. I would have liked to see a comparison of uncertainty estimation and anomaly detection on the main task. For instance, one could take the task of quality assessment on a production line by computer vision. The model would output an uncertainty score for each prediction. Predictions with high uncertainty are most likely anomalies. How does this compare to anomaly detection as binary classifiers? I think a discussion of this type could improve the paper as it would clarify this kind of doubt in other people like me reading the paper.

More weaknesses below in the requested changes.

---

> ### Author Response · Authors · 2024-12-12
> **Response to reviewer FR6C**
>
> Dear reviewer FR6C,
>
> We appreciate the detailed feedback and thoughtful suggestions, which have provided valuable insights to strengthen our work. We have fixed the text and marked all changes in orange.
>
> Below, we address the primary concerns and proposed improvements.
>
> - [**Motivation for Binary Classification in Anomaly Detection**] While anomalies can indeed be viewed as marked outliers relative to a data distribution, anomaly detection is often treated as a binary classification task [Chandola et al. (2009), Ruff et al. (2019), Fung et al. (2023)]. This is especially useful for real-world applications where (i) anomalies need to be flagged explicitly for downstream decisions (e.g., defective vs. non-defective items in quality control), or (ii) a model’s epistemic uncertainty is insufficient to distinguish anomalies from highly noisy in-distribution samples. To clarify, uncertainty estimation alone may not always identify anomalies effectively. For instance, predictions with high uncertainty could correspond to samples near class decision boundaries (high aleatoric).
>
> - [**Sampling $N$ Labels in Section 3.2**] The term $P(Y=1|X)$ can be understood as the proportion of anomalous labels obtained when collecting labels for the same instance, similar to crowdsourcing. For example, given an image $X$, how many labelers would classify it as anomalous? While a figure could help, we believe that refining the explanation with additional text may maintain focus on the narrative.
>
> - [**Clarifying Eq. 3 Assumptions**] We acknowledge that Eq. 3 relies on the assumption that the rarity score can approximate data density. However, one can transform any data similarity/dissimilarity metric into a weak estimator for the data density [1] and the rarity score fits in this category. In addition, we want to highlight that our rarity-based weak estimator is enough for our task for two reasons. First, assigning null density to synthetic anomalies falling outside of all training spheres allows unrealistic anomalies to be detected. Second, the provided implementation is fast, as one can compute the training spheres only once for the whole set of synthetic examples. We include this in the paper.
>
> - [**$f(x)$ in Eq.4**] The notation $f(x)$ indicates the anomaly score assigned by the anomaly detector, as specified in Sec. 3.1, right after Def. 3.1. Because Eq.4 transforms the anomaly scores into class conditional probabilities, we require indicating such scores.
>
> - [**Discussing P3 and Anomaly Density Trade-Offs**] Our method does indeed depend on observed anomaly density, but also on the density of the normal samples. In fact, P3 depends on the whole data density (not class) and can be read as follows: assuming that the synthetic anomaly is distinguishable for the anomaly detector from a normal counterpart, the closer to the training instances (normal or anomalous) the higher the quality. Roughly speaking, such an anomaly would improve the decision boundary when used for training.
>
> - [**Experimental setup for synthetic anomalies**] We use $l/3$ of each type of anomaly to construct our synthetic dataset, while the original training set contains only real anomalies. Because we rely on ranking-based metrics like AUROC to evaluate scoring functions, our results remain reliable even with small proportions of realistic anomalies.
>
> - [**Q5, Table 3**] We move Table 3 to the main body of the paper and include the AUC values for completeness.
>
> - [**Avoid excessive parentheses and explain rAUC**] We revise the section, simplify the text, and provide further details for the rAUC metric.
>
> We hope this addresses your concerns. Thank you again for your constructive feedback, which has been helpful in refining our work.
>
> [1] Breunig MM, Kriegel HP, Ng RT, Sander J. LOF: identifying density-based local outliers. ACM SIGMOD, 2000.

---

### Review · Reviewer_s2gF · 2024-11-28

**Summary Of Contributions:**

1. This work proposes expected anomaly posterior (EAP), an uncertainty-based score function that measures the quality of auxiliary anomalies.
2. The authors analyze theoretical properties of EAP.
3. The authors show that EAP outperforms 12 adapted data quality estimators on 40 benchmark datasets.

**Audience:**

Yes

**Broader Impact Concerns:**

I don't have any concerns.

**Claims And Evidence:**

Yes

**Requested Changes:**

Please refer to the weaknesses above.

**Strengths And Weaknesses:**

Strengths
1. Measuring the quality of auxiliary anomalies is an important task, considering that labeled anomalies are hard to acquire.
2. The proposed method shows strong empirical performance on various datasets, compared to many competitors.
3. The authors have performed various experiments, including zero-shot anomaly detection, to evaluate their approach.

Weaknesses
1. In Definition 3.1, it seems incorrect to define unrealistic anomalies as having null density, since every sample in the data space can have non-zero density even though it is highly unrealistic. Can you provide any concrete examples or logic to support your choice of Definition 3.1, based on real-world scenarios?
2. The problem setup is confusing. Anomalies are usually defined as data samples that are different from the observed training samples. Then, “$p(X)$ is low” is the same as “$P(Y=1|X)$ is high.” as the authors also mention as “epistemic uncertainty tends to be high for most anomalies.” Is it correct to consider $p(X)$ and $P(Y=1|X)$ separately, treating anomalies like a different label?
3. The conditional probability $\hat{P(Y=1|X=x)}$ that the authors aim to estimate in Eq. (4) is the same as $P(Y|x)$, which is the goal of anomaly detection. If one can successfully estimate the probability, why not directly solving the problem?
4. I can’t understand the intuition behind Eq. (4), which is an essential part of the proposed method. A more detailed explanation would be helpful.

---

> ### Author Response · Authors · 2024-12-12
> **Response to reviewer s2gF**
>
> Dear reviewer s2gF,
>
> We sincerely thank you for your feedback, which has greatly helped us improve our work. We have fixed the text and marked all changes in orange.
>
> Below, we address each concern in detail:
>
> - [**Unrealistic anomalies and null density**] We agree that all samples in a continuous data space may have non-zero density under any probabilistic model within its support. However, our definition of “unrealistic anomalies” refers to samples falling **outside the support of the real data generation model**. These are not merely low-density samples, which can indeed be interesting anomalies, as anomalies are naturally rare and often occur in low-density regions of the real data distribution. Instead, unrealistic anomalies exist fundamentally outside the distributional space of plausible data. For instance, consider an image of a bottle when the training set includes only normal and broken toothbrush images. Such a sample lies in a zero-density region of the real data generation model because it does not share any semantic similarity with the training data. Please refer to Sec 5.1 (Data) for details on how we introduce unrealistic anomalies in the experiments.
>
> - [**Separation of $p(X)$ and $p(Y=1|X)$**] Anomaly detection is often treated as a (un/supervised) binary classification task, where anomalies belong to a specific class [Chandola et al. (2009), Vercruyssen et al. (2018), Ruff et al. (2019), Fung et al. (2023)]. In this context, the distinction between $p(X)$ and $p(Y=1|X)$ is crucial: $p(X)$ reflects the density of the region where a sample lies, capturing how typical or atypical the sample is relative to the overall data distribution, while $p(Y=1|X)$ leverages available label information to quantify how confident a discriminative model is in classifying a sample as anomalous. This distinction allows for a thorough evaluation of anomalies, considering both distributional divergence (via $p(X)$) and classification confidence (via $p(Y=1|X)$). However, it is important to note that neither metric alone is sufficient to assess anomaly quality comprehensively. This limitation is evident in our experiments, where we demonstrate the advantages of combining these perspectives.
>
> - [**Estimating $P(Y=1|X)$ solves the problem**] We agree that if one could have a calibrated reliable estimator for $P(Y=1|X)$, the task would be solved. Unfortunately, this is challenging in anomaly detection as highlighted in Sec. 3.2. Thus, our framework employs a weak estimator for $P(Y=1|X)$, which requires the introduction of $p(X)$ to properly evaluate the quality of anomalies.
>
> - [**Details on Eq 4**] Eq.4 transforms the real-valued anomaly scores $f(x)$ to probabilities in $[0,1]$ using a monotonic function. Given a decision threshold $\lambda$, one wants to map its value to $0.5$, all greater values to $(0.5, 1]$, and all lower values to $[0,0.5)$ such that, when transforming the probabilities to predictions by thresholding at $0.5$, the predicted classes are maintained. This transformation works better than the simple linear and Gaussian, as shown in Vercruyssen et al (2018).

---

### Decision · Action_Editor_N1Nk · 2024-12-23

**Recommendation:** Accept as is

**Comment:**

The 3 reviewers recommend accept, leaning accept and leaning reject.  While not all recommendations are "accept," the task of evaluating auxiliary/synthetic anomalies is interesting, which might benefit the community.   Two of the reviewers are satisfied with the revision and the third didn't indicate additional revisions.

**Audience:**

Part of the TMLR's audience would find this article interesting, particularly the anomaly detection community.

**Claims And Evidence:**

The authors propose expected anomaly posterior (EAP) to evaluate the quality of auxiliary/synthetic anomalies, which can be used to help train an anomaly detector.  The authors assume a Bernoulli distribution over the data and model the probability of sampling an anomaly out of N (n*p(x)) samples of the same x as the conditional probability P(anomaly|x), for which the expectation (quality score) is computed when N is high.

The authors evaluated their proposed EAP against 12 existing methods over 40 datasets.  Evaluation includes the ranking of anomalies and performance of downstream tasks. Empirical results indicate that EAP generally outperforms the existing methods included in the experiments.